# Cracking Resistance of Recycled Rubber Asphalt Binder Composed of Warm-Mix Additives

**DOI:** 10.3390/ma15134389

**Published:** 2022-06-21

**Authors:** Wanmei Gui, Li Liang, Lan Wang, Fei Zhang

**Affiliations:** 1Resources and Civil Engineering, Northeastern University, Shenyang 110000, China; guiwanmei_0620@126.com (W.G.); ll-neu@163.com (L.L.); 2Civil Engineering, Inner Mongolia University of Technology, Hohhot 010051, China; 20210000034@imut.edu.cn

**Keywords:** simple fractional model, thermal stress, critical cracking temperature, warm-mixed crumb rubber-modified asphalt binder

## Abstract

Warm-mix asphalt technology has been applied to recycled rubber asphalt binder (RAB), which forms warm-mixed crumb rubber-modified asphalt binder (W-RAB) as a “green” material for environmental conservation and to enhance road performance. Furthermore, low-temperature cracking is one of the major distresses for asphalt pavement, which drastically restricts ride quality and service level. Therefore, the main objective of this study is to comparatively analyze the low-temperature properties of W-RABs based on thermal stress and the simple fractional model. W-RABs were obtained by mixing 60 mesh recycled rubber (CR) and two different types of warm-mix additives, namely viscosity reducer (1, 2, and 3%) and surfactant (0.4, 0.6, and 0.8%). First, Hopkins and Hamming’s numerical algorithm and the Boltzmann superposition principle were used for obtaining thermal stress $\sigma \left(T\right)$. Subsequently, critical cracking temperature $T_{cr}$ was derived using the single asymptote procedure (SAP) theory. Second, the simple fractional viscoelasticity model was used to calculate the creep compliance, damping ratio, and dissipation energy ratio, and the results were compared with the Superpave protocol results obtained with bending beam rheometer (BBR) tests. The results showed that a combination of CR and warm-mix additives could slightly improve the thermal crack resistance of the asphalt binder. The addition of 0.6% surfactant yielded the optimum performance, while only a high dosage (3%) of viscosity reducer provided a marked improvement in efficiency, which decreased with a decrease in temperature. This study recommends the use of RAB composited with 0.6% surfactant for areas with extremely low temperature.

## 1. Introduction

### 1.1. Background

Low-temperature cracking is one of the major distresses found in asphalt pavement built under low-temperature climatic conditions. A significant increase in thermal stress accumulation ultimately results in the generation of transverse cracks extending at the surface of the pavement when the temperature drops below a certain limit value known as the critical temperature [1,2]. Asphalt binder plays a dominant role in the low-temperature performance of asphalt mixture, contributing up to 80% according to the Strategic Highway Research Program (SHRP) [3]. Therefore, it is essential to use thermal stress as an evaluation criterion to predict low-temperature behaviors of asphalt binder, and apply green road construction technology to improve its road performance. One example is combining recycled rubber (CR) asphalt binder (RAB) with warm-mix additive.

The utilization of RAB is an effective way of consuming scrap tire rubber and enhancing road performance, such as improved resistance to cracking and permanent deformations. It was reported that temperature susceptibility of RAB can be decreased at low-temperature zones by decreasing the CR content, which can diminish stiffness and elasticity [4]. However, this leads to the emission of a large amount of toxic and harmful gases, such as greenhouse gases, creating environmental pollution. For instance, Pouranian et al. [5] showed that RABs released some hazardous fumes, such as acetone and meta/para-xylene, through gas chromatography–mass spectrometry analysis. Yang et al. [6] found that asphalt mixtures containing CR primarily released pollutants such as xylene and toluene. For these reasons, the warm-mix asphalt (WMA) technology was applied to reduce harmful emissions, reduce energy consumption, and improve construction workability by reducing the viscosity of the asphalt binder. In general, this technology can be subdivided into three main categories, namely organic wax-based, foaming-based, and chemical surfactant-based technologies [7,8,9,10,11]. Ma et al. [12] found that both solid polyolefin additive and liquid surfactant could lower the compaction temperatures of rubber asphalt mixtures by approximately 10–20 °C, and showed no adverse effects on the low-temperature performance of rubber asphalt mixtures. Luo et al. [13] developed a novel three-component WMA additive, which was a silicon-based quaternary ammonium salt surfactant, prepared using silica gel as the main component. Notably, this additive did not affect asphalt binder performance at pavement service temperature. Several studies have been carried out on the addition of WMA additives to RABs focusing on the determination of the CR content, CR particle size, WMA additive content, WMA additive type, as well as their influence on asphalt binder performance.

Bending beam rheometer (BBR) tests are universally used to obtain mechanical parameters, such as creep stiffness $S\left(t\right)$, creep rate m, and creep compliance $D\left(t\right)$, which can be used to evaluate the low-temperature properties of asphalt binders, according to ASTM 2016 [14]. Direct tension tester (DTT) tests are considered to be the most promising approach to predict the low-temperature properties of asphalt mixtures, but they require a very sensitive machine, which cannot be widely supported by manufacturers [15]. Moreover, depending on the approximate and analytical interconversion solutions, the thermal stress $\sigma \left(T\right)$ can be determined by interconverting $S\left(t\right)$ with the corresponding relaxation modulus. The common interrelationships include the use of power law or two exact numerical algorithm procedures, such as the convolution integral based on Hopkins and Hamming’s algorithm and the Laplace transformation [16]. The critical cracking temperature $T_{cr}$ can be subsequently derived from the single asymptote procedure (SAP) without the need for more sophisticated strain-controlled relaxation modulus test [17]. Moreover, Xu et al. [18] demonstrated that $\sigma \left(T\right)$ and $T_{cr}$ could be used to effectively analyze the influence of WMA additives (Sasobit and Et-3100) on the low-temperature properties of asphalt binder materials.

However, the aforementioned test methods are time-consuming. Therefore, more effective viscoelastic mechanical models or equations have been implemented to suitably predict the low-temperature performance of asphalt binders over a wide range of temperatures and frequencies [19]. The Burgers model is the most commonly implemented method and is composed of the following four components: two spring and two dashpots with four parameters [20,21]. Furthermore, a fractional calculus element, called the spring-pot element, was proposed to characterize the mechanical property of the viscoelastic material which is transformed from a solid state to a fluid. Consequently, some researchers have recently resorted to the simplest real-order model containing a spring-pot element because of its sufficient precision and fewer parameters compared with the integer order models. Hajikarimi et al. [22,23] showed that simple fractional model with general power law could be suitably substituted for the Burgers model or the generalized Maxwell model, which was then used to determine the low-temperature characteristics of modified asphalt binders with polyphosphoric acid (PPA) and distillate aromatic extracts (DAEs) of oil.

Most previous studies have used trends of $S\left(t\right)$ and m values based on BBR tests and traditional viscoelastic models to evaluate the effect of incorporating WMA additives on the low-temperature performance of asphalt binders. However, few studies have explored the thermal stress $\sigma \left(T\right)$ and the corresponding critical cracking temperature $T_{cr}$ of modified asphalt binders.

### 1.2. Research Objective

In the present study, the effects of types and dosages of WMA additives on the low-temperature performance of RABs were investigated, and the optimum was recommended for the cold region of Inner Mongolia, China. To achieve the research objectives, three main approaches were adopted. (1) Hopkins and Hamming’s numerical algorithm was applied to determine the relaxation modulus and corresponding rates based on the results of BBR tests. Then, model master curves were constructed using the Christensen–Anderson–Marasteanu (CAM) model. (2) The Boltzmann superposition principle was used for obtaining thermal stress $\sigma \left(T\right)$. Subsequently, critical cracking temperature $T_{cr}$ was derived using the SAP theory. (3) The derivation of creep compliance, damping ratio, and dissipated energy ratio was analyzed using the parameters $\left(A,\;a\right)$ obtained from the simple fractional model. Finally, the most suitable dosage and type of WMA additive were selected to obtain the best modified asphalt binder for frigid regions.

## 2. Materials and Experimental Methods

### 2.1. Materials and Sample Preparation

In this study, a penetration grade 80–100 virgin binder from the northwest region of China was used to blend modifiers. CR powder (60 mesh), viscosity reducer LP (white flaky solid), and surfactant additive SK (yellow–brown emulsion) were supplied by the Transportation Research Institute, China. The dosages (by weight of binder) of CR powder, LP additive, and SK additive were 15–20, 0.8–3, and 0.25–0.75%, respectively, as recommended by the suppliers and previous research [24,25,26,27]. Therefore, in this study, the dosage of each modifier was selected as follows: CR powder 20% (60 mesh); LP additive at 1, 2, and 3%; and SK additive at 0.4, 0.6, and 0.8% using the mass of the base binder, respectively. The related images of WMA additives are shown in Figure 1. The technical specifications of CR based on the requirements of “Road Waste Vulcanized Rubber Powder” (JT/T 7997-2011) are presented in Table 1.

### 2.2. Preparation of the Composited Modified Asphalts

First, 20% of 60 mesh CR powder was mixed with base asphalt binder for 30 min at 180 °C to make the CR completely grow or swell in the asphalt binder. Consequently, the RAB was obtained. Second, predetermined amounts of LP and SK were separately added to RAB under the same conditions to ensure homogeneous mixing, and the contents were mixed for about 15 min. Finally, one RAB and six W-RABs (S0.4-RAB, S0.6-RAB, S0.8-RAB, L1-RAB, L2-RAB, and L3-RAB) were obtained. The nomenclature of asphalt binders is presented in Table 2.

### 2.3. Test Method

Based on ASTM D6521 [28], all samples were subjected to the short-term aging procedure after placing them in a rolling thin film oven (RTFO) and keeping them at 163 ± 0.5 °C. Bending beam rheometer (BBR) tests were performed on thin asphalt binder beams (102 ± 5 mm×12.7±0.5 mm×6.25±0.5 mm) after applying the creep load at the middle for 240 s, according to the ASTM 2008 specification [4]. The tests were conducted at four different performance grade (PG) temperatures $\left(-12,-18,-24,-30\;{}^{\circ}C\right)$, and the mid-span deflection values, $\delta \left(t\right)$, were recorded every 0.5 s. Three replicate specimens for each W-RAB were tested at each temperature in this study. Various parameters based on $\delta \left(t\right)$, such as creep stiffness, $S\left(t\right)$, creep rate, and $m\left(t\right)$, were calculated. The applied research approach is shown in Figure 2.

## 3. Theoretical Basis

### 3.1. Superpave Protocol

The $S\left(t\right)$ and $m\left(t\right)$ values were calculated using the experimental results of mid-span deflection $\delta \left(t\right)$, which can be used to evaluate the low-temperature performance, as follows:(1)$$\;S\left(t\right)=\frac{\sigma }{\varepsilon \left(t\right)}=\frac{{\mathrm{Pl}}^{3}}{{4\mathrm{bh}}^{3}\delta \left(t\right)}=\frac{1}{D\left(t\right)}$$
(2)$$\;m\left(t\right)=\left|\frac{\mathrm{dlogS}\left(t\right)}{\mathrm{dlog}\left(t\right)}\right|$$
where $S\left(t\right)=$ the creep stiffness; $D\left(t\right)=$ the creep compliance; $\sigma \left(t\right)=$ the maximum bending stress; $m\left(t\right)\;$= the creep rate; P= the constant applied load $\left(980\pm 50\;\mathrm{mN}\right)$; and l=102±5 mm, b=12.7±0.5 mm, and h=6.25±0.5 mm are the length, width, and height of specimen, respectively.

### 3.2. Thermal Stress and Critical Cracking Temperature

#### 3.2.1. Thermal Stress

The low-temperature pavement performance of RAB was evaluated using thermal stress $\sigma \left(\xi \right)$ according to the current AASHTO Mechanistic–Empirical Pavement Design Guide (MEPDG) [29]. In this study, the parameter $\sigma \left(\xi \right)$ was calculated using the following steps:(a)The creep compliance $D\left(t\right)$ was obtained from the BBR test results of $S\left(t\right)$, $D\left(t\right)=1/S\left(t\right),$ as given by Equation (1).(b)Then, $D\left(t\right)$ was converted to the corresponding relaxation modulus $E\left(t\right)$ by solving the convolution integral, as shown in Equation (3), after applying the Hopkins and Hamming algorithm, as given by Equation (4).(c)The $E\left(t\right)$ master curve was drawn using the BBR experimental data obtained at four different temperatures and the Christensen–Anderson–Marasteanu (CAM) model, as shown in Equation (5).(d)Then, $\sigma \left(\xi \right)$ was calculated by solving the one-dimensional hereditary integral, as given by Equation (7).


(3)$$\int_{0}^{t}D\left(t\right)E\left(t-\tau \right)d\tau =t$$(4)$$\begin{array}{cc} E\left(t_{n+1}\right) & {=(t}_{n+1}-\overset{n-1}{\underset{i=0}{\sum }}\left({E(t}_{i+1/2}\right)\left[f\left(t_{n+1}{-t}_{i}\right)-f\left(t_{n+1}{-t}_{i+1}\right)]\right)/{(f(t}_{n+1}-t_{n}))\\ & {=(t}_{n+1}-\overset{n-1}{\underset{i=0}{\sum }}\left({E(t}_{i+1/2}\right)[f(t_{i+1}{)-f(t}_{i}){])/(f(t}_{n+1}-f\left(t_{n}\right)) \end{array}$$
where t= the time interval $(t_{0}=0,\;t_{1}=1,\;t_{2}=2,\ldots ,\;t_{240}=240$ and $t_{i+1/2}=\left(t_{i+1}+t_{i}\right)/2)$. The initial value f($t_{0})=$ 0, $E\left(t_{0}\right)=$ 0, $E\left(t_{1}\right)=t_{1}/f(t_{1})$.
(5)$$\;E\left(t\right){=E}_{g}{\left[1+{\left(\frac{t}{t_{c}}\right)}^{v}\right]}^{-\frac{w}{v}}\;\Rightarrow \mathrm{logE}\left(t\right)=-\frac{w}{v}\log\left\{1+{\left[10^{\log\left(t\right)+\log\left(a_{T}\right)-\log\left(t_{c}\right)}\right]}^{v}\right\}$$
where $t_{c},\;v,\;w=$ the fitting parameters; $E_{g}=$ the glassy modulus $\left(3\;\mathrm{GPa}\right)$; and $a_{T}=$ the horizontal shift factor, which can be expressed using the Williams–Landel–Ferry (WLF) equation as follows [2,16]:(6)$${\;\mathrm{loga}}_{T}=\frac{{-C}_{1}\cdot \left({T-T}_{\mathrm{ref}}\right)}{C_{2}+\left({T-T}_{\mathrm{ref}}\right)}$$
where $C_{1},\;C_{2}=$ the fitting parameters; T= the test temperature (°C); and $T_{0}=$ the reference temperature (−12 °C).
(7)$$\sigma \left(\xi \right)=\int_{-\infty }^{\xi }\frac{d\varepsilon \left(\xi^{\prime }\right)}{{d\xi }^{\prime }}E\left({\xi -\xi }^{\prime }\right){\;d\xi }^{\prime }\;=\int_{-\infty }^{t}\frac{d\left(a△T\right)}{d\left(t^{\prime }\right)}\cdot E\left(\xi \left(t\right){-\xi }^{\prime }\left(t\right)\right){d(t}^{\prime })$$
where aΔT= the thermal strain; ΔT= the temperature variation $\left(+20–40\;{}^{\circ}C\right)$ with cooling rate (v) =0.2, 1, 5, 20 °C/h; a= the thermal expansion coefficient (0.00017); and $\xi \left(t\right)=\int_{0}^{t}\frac{dt^{\prime }}{a_{T}}$.

#### 3.2.2. Critical Cracking Temperature

The index critical cracking temperature $T_{CR}$ can be estimated from the $\sigma \left(\xi \right)$ curve according to SAP proposed in a literature study [17]. Shenoy’s theory used only the BBR test results to calculate $T_{CR}$ and the results correlated well with the data from the direct tension test (DTT) [20]. The SAP process is depicted in Figure 3, where the *Y*-axis represents the $\sigma \left(\xi \right)$ and the *X*-axis represents the temperature. The asphalt temperature–stress curve during the cooling process first rises slowly and finally exhibits a sharp rise, and the asymptote lines at the beginning and end of the curve represent the limit curvature of the temperature stress accumulation, respectively. The parameter *T_CR_* (°C) is the intersection between the *X*-axis and its asymptote, which is calculated by fitting the thermal stress curve at the low-temperature part using the Origin software. Thus, in this study, $T_{CR}$ was obtained using the SAP theory to assess the effect of W-RAB on the low-temperature crack resistance. The higher the $T_{CR}$ value, the greater the possibility of asphalt binder cracking.

#### 3.2.3. Simple Fractional Viscoelastic Model

According to the literature [30], a fractional calculus element indicated by the general power law was proposed, which is called the spring-pot element, as shown in Figure 4. The constitutive equations, creep compliance $D\left(t\right),$ and relaxation modulus $E\left(t\right)$ of simple fraction element can be, respectively, defined as follows [22]:(8)$$\;\sigma \left(t\right){=E\tau }^{a}D^{a}\varepsilon \left(t\right),\;\left(0\leq a\leq 1\right)$$
(9)$$\;D\left(t\right)=\frac{\varepsilon \left(t\right)}{\sigma }{=\mathrm{At}}^{a}$$
(10)$$\;E\left(t\right)=\frac{t^{-a}}{A\Gamma \left(1+a\right)\Gamma \left(1-a\right)}$$

By implementing a Fourier transformation of Equation (10) and converting the time domain to the frequency domain, the storage modulus $E^{\prime }$ and the loss modulus $E^{\prime \prime }$ can be derived as follows:(11)$$E^{\prime }\left(w\right)=\frac{w^{a}}{A\Gamma \left(1+a\right)}\cos\left(\frac{\pi a}{2}\right)$$
(12)$$E^{\prime \prime }\left(w\right)=\frac{w^{a}}{A\Gamma \left(1+a\right)}\sin\left(\frac{\pi a}{2}\right)$$

Creep stiffness $S\left(t\right)$ can be obtained using Equation (9), which is expressed as a reciprocal relationship to $D\left(t\right)$, as follows:(13)$$\;S\left(t\right)=\frac{1}{D\left(t\right)}=\frac{1}{A}t^{-a}$$

Then, by applying the algorithm on both sides of Equation (13), the following equation can be obtained:(14)$$\;\log\;S\left(t\right)=-\log\;A-\mathrm{alog}\;\left(t\right)$$

By combining Equations (2) and (14), it can be simply proven that m-value depends only on the power of a and is independent of time, as follows: (15)$$\;m=\left|\frac{\mathrm{dlogS}\left(t\right)}{\mathrm{dlog}\left(t\right)}\right|=\left|\frac{d\left\{-\mathrm{logA}-\mathrm{alog}\left(t\right)\right\}}{d\left\{\log\left(t\right)\right\}}\right|=a$$

The damping ratio is defined as Equation (11) divided by Equation (12), which characterizes the inherent property of material resistance to deformation, as follows:(16)$$\;\mathrm{Damping}\;\mathrm{Ratio}=\frac{E^{\prime \prime }\left(w\right)}{E^{\prime }\left(w\right)}=\tan\left(\frac{\pi a}{2}\right)$$

The dissipated energy ratio $\left(DER\right)$ can be calculated as the ratio of dissipated energy $W_{d}\left(t\right)$ to the stored energy $W_{s}\left(t\right)$ in time domain, in order to characterize the ability of asphalt binder to dissipate energy, as follows: (17)$$W_{d}\left(t\right){=A\sigma }^{2}\left[\frac{1}{2}{\left(2t\right)}^{a}\right]$$
(18)$$W_{s}\left(t\right){=A\sigma }^{2}\left[t^{a}-\frac{1}{2}{\left(2t\right)}^{a}\right]$$
(19)$$\mathrm{DER}=\frac{2^{a-1}}{{1-2}^{a-1}}$$
where A, a= the constant parameters of the spring-pot element, $\Gamma \left(t\right)=$ the gamma function, and $\Gamma \left(t\right)=\int_{0}^{\infty }x^{t-1}e^{-x}dx$.

## 4. Results and Discussion

### 4.1. Thermal Stress and Critical Cracking Temperature

#### 4.1.1. Relaxation Modulus *E*(*t*)

As described in Section 3.2.3, the Hopkins and Hamming interconversion algorithm was selected for estimating the relaxation modulus $E\left(t\right)$, as given by Equation (4). Figure 5 shows that the curves of $E\left(t\right)$ present a downward trend over time. Moreover, the RAB showed larger $E\left(t\right)$ values than the remaining asphalt binders over most of the time range. Moreover, the curves of L-RAB with different contents of LP were almost parallel and showed an almost constant slope of the curve, whereas a 3% dosage of LP showed the lowest $E\left(t\right)$ values (Figure 5a,b). A similar change was observed for S-RAB with different contents of SK, as shown in Figure 5c,d, where the $E\left(t\right)$-value of S0.6-RAB was the lowest. Clearly, both the type and the dosage of the WMA additive played a key role in determining the relaxation characteristics of the modified asphalt binder.

#### 4.1.2. Master Curves for $E\left(t\right)$
and $m_{E}$

In order to calculate the thermal stress $\sigma \left(\xi \right)$, the master curves of $E\left(t\right)$ were first obtained (Section 3.2.3). Figure 6 illustrates the $E\left(t\right)$ movement process for L3-RAB in a log–log scale as an example. The $E\left(t\right)$ curves plotted at −12, −18, −24, and −30 °C became horizontally shifted to the reference temperature. Eventually, the $E\left(t\right)$ master curve of L3-RAB was obtained at −12 °C. Furthermore, Figure 6 presents that the $E\left(t\right)$ master curve trend could be effectively predicted using the CAM model. Figure 7 displays the master curves for $E\left(t\right)$ drawn at −12 °C for seven W-RABs. The master curve of $E\left(t\right)$ is helpful to evaluate the low-temperature performance of modified asphalts, because it can comprehensively reflect the deformability and relaxation ability over a wide range of temperature and time domain. Of note, Figure 7 exhibits that the master curve of RAB is located above the other three curves, which indicates that it shows the largest $E\left(t\right)$ over a wide temperature range. Moreover, it was also observed that the effects of various dosages of WMA additives were different. For example, L3-RAB and S0.6-RAB showed significantly smaller $E\left(t\right)$ values. In general, the smaller the value of $E\left(t\right)$, the better the low-temperature performance. Therefore, both WMA additives showed a positive effect on low-temperature resistance, and the impact of dosage played a dominant role.

Figure 8 presents the curve of slope $m_{E}$ of *log E*(*t*) versus *log* (*t*), which was simply calculated using the first derivative of the corresponding fitted models. Clearly, the $m_{E}$-value in the lower creep period approached zero; however, the stress relaxation speed gradually increased with an increase in the creep period. The comparative analysis indicated the occurrence of a remarkable difference during the last creep period. S0.6-RAB showed the maximum $m_{E}$-value and RAB showed the lowest. Lower $m_{E}$-values are undesirable because of the lower relaxation rate. The above-mentioned results are consistent with the conclusion presented in the previous section, which stipulates that S0.6-RAB exhibited the best performance. Therefore, this result reveals that $m_{E}$ can validly represent the thermal anti-cracking ability of modified asphalt binders.

#### 4.1.3. Thermal Stress $\sigma \left(T\right)$

Thermal stress $\sigma \left(T\right)$ evolution is directly related to the relaxation modulus. In this study, $\sigma \left(T\right)$ values of all W-RABs were calculated based on the Boltzmann superposition principle, and the related computation processes are explained in Section 3.2.3. Figure 9 presents the thermal stress curves for all asphalt binders at different cooling rates (0.2, 1, 5, 20 °C/h). An identical trend was observed for all asphalt binders. $\sigma \left(T\right)$ increased slowly alongside a decrease in temperature, and initially the curve remained almost constant, but then increased rapidly when the temperature dropped from −30 to −40 °C. Therefore, a significant increase in the thermal stress accumulation occurred, which ultimately resulted in the extension of the transverse cracks at the surface of the pavement when the temperature dropped below the critical temperature. Another important trend is that an increase in the cooling rates gradually increased thermal stress accumulation. In particular, the $\sigma \left(T\right)$ was nearly three times higher when the rate increased to 20 °C/h, thus degrading the thermal anti-cracking ability.

Moreover, Figure 9 shows that $\sigma \left(T\right)$ at different cooling rates presented the same variation tendency with different dosages and types of WMA additives. Notably, when the $\sigma \left(T\right)$ values were influenced solely by SK, the anti-cracking capability of S-RABs in descending order is as follows: S0.6-RAB > S0.8-RAB > S0.4-RAB, while the order of L-RABs is L3-RAB > L2-RAB > L1-RAB. As depicted, the modification ability of composite modifiers was greater than that of the single modifier alone. Comprehensive comparative analysis indicates that both types of WMA additives could optimize the thermal crack resistance; however, the optimum addition was 0.6% SK as S0.6-RAB exhibited the lowest $\sigma \left(t\right)$ value. It is likely that SK forms a silane coupling layer to prevent asphaltene from coagulation and thus reduces its viscosity, leading to improved low-temperature properties [13]. In contrast, very little difference was observed between RAB and L1-RAB. Only a high dosage of LP could significantly improve the corresponding behavior to compensate for the weakness of L-RABs at low temperatures.

The results of critical cracking temperature $T_{CR}$ were obtained using the SAP (Section 3.2.3), as presented in Table 3. With an increase in the cooling rate, gradual growth was observed in $T_{CR}$. In particular, $T_{CR}$ increased by approximately 5 °C following an increase in the rate from 0.2 to 20 °C/h (Table 3), indicating greater potential to crack. In terms of the influence of WMA additives, W-RABs exhibited lower $T_{CR}$ value than RAB, S-RABs exhibited a lower $T_{CR}$ value than L-RABs, and S0.6-RAB showed the minimum value. Therefore, the most effective approach was to add 0.6% SK to RAB, which led to improvements in its low-temperature property, consistent with the analysis of $\sigma \left(T\right)$.

### 4.2. Simple Fractional Viscoelastic Model Analysis

#### 4.2.1. Determination of A, a-Value for the Simple Fractional Model

Two different model parameters A, a associated with the simple fractional viscoelastic model were calculated from Equations (1) and (13)–(15) based on the nonlinear fitting of BBR experimental data using MATLAB 7.12. According to Equation (15), it is simply shown that a constant m-value is only related to the power of a, because the changing trend of spring-pot $D\left(t\right)$ with time in the log–log plot is a straight line, causing a reciprocal relationship between $D\left(t\right)$ and $S\left(t\right)$. The m-values calculated from the Superpave protocol and the model are shown in Figure 10.

Figure 10 presents the two m-values determined by different methods, which were in good agreement because all data points were clustered around the identical line. This indicates that the fractional viscoelastic model can be used to rationally analyze the rheological and mechanical behavior of modified asphalt binders. The A, a results of all samples at four different temperatures (−12, −18, −24 and −30 °C) are summarized in Table 4. The results indicate that a decrease in the temperature decreased the corresponding a and A values, which indicates worse low-temperature anti-cracking properties. Irrespective of the amount of WMA additives added to modify RAB, values of A and a showed an increment. Evidently, the addition of WMA additives showed a positive impact on the low-temperature properties of RAB.

#### 4.2.2. Creep Compliance $D\left(t\right)$ and Derivation of Creep Compliance $D^{\prime }\left(t\right)$

Aflaki et al. [31] and Hajikarimi et al. [22] also proposed a comprehensive evaluation indicator, namely the derivation of creep compliance $D^{\prime }\left(t\right)$, to describe the low-temperature properties of modified asphalt binders. This method can avoid conflicting situations with a single indicator, such as the Superpave protocol indicator S or the m value. Both $D\left(t\right)$ and $D^{\prime }\left(t\right)$ can be determined using the values of A and a, as presented in Table 4. In the fractional viscoelastic model, a higher value of $D\left(t\right)$ or $D^{\prime }\left(t\right)$ indicates better low-temperature rheological properties, which can be calculated using Equations (9) and (20), respectively:(20)$$D^{\prime }\left(t\right)=\frac{m\left(t\right)}{S\left(t\right)}\times \frac{1}{t}\;\approx {\;\mathrm{aAt}}^{-a-1}$$

In order to assess the accuracy of the fractional model, the model and experimental $D\left(t\right)$ results at different test temperatures (−12, −18, −24 and −30 °C) for RABs were obtained, as presented in Figure 11. Clearly, the majority of the data fell either on the line of equality or in the nearby regions. The statistical parameter $R^{2}$ value was almost 0.99, which further indicates that the general power law can be used to analyze the creep performance of asphalt binders.

The fractional model results of $D\left(t\right)$ over the entire creep period and $D^{\prime }\left(t\right)$ at 60 s for W-RABs at various temperatures are summarized in Figure 12 and Figure 13, respectively. The results indicate that by decreasing the test temperature, the values of both D and $D^{\prime }\left(t\right)$ decreased, and exhibited a decreasing viscous behavior that led to less stress relaxation capability. Furthermore, at the same low temperature, the incorporation of WMA additives in RAB led to a significant improvement in the results of $D\left(t\right)$ and $D^{\prime }\left(t\right)$. However, there were slight differences between W-RABs with different dosages. For the S-RABs, values of both indicators improved by approximately 30% when the SK dosage increased to 0.6%. For the L-RABs, with an increase the in LP content from 1 to 3%, the corresponding values increased gradually. Nonetheless, these values were lower than those of S-RABs, especially for L1-RAB and L2-RAB. Therefore, it was found that test temperature, WMA additive content, and interaction between the WMA additive and CR significantly affected the values of D and $D^{\prime }\left(t\right)$. Both LP and SK additives showed a slight positive impact on the low-temperature properties of RAB, with the most remarkable improvement found for S0.6-RAB.

#### 4.2.3. Damping Ratio of Asphalt Binders

The damping ratio indicates the inherent resistance to deformation [32]. Equation (16) indicates that the damping ratio is the ratio of the loss modulus $E^{\prime \prime }\left(w\right)$ to the storage modulus $E^{\prime }\left(w\right)$ and is related only to the power a, which indicates that it is only related to the m-value. Figure 14 presents the damping ratio values for all W-RABs versus temperatures, calculated using Equation (16).

First, it is evident from the plot that the damping ratio for all asphalt binders proportionally decreased with a decrease in the test temperature. Such a response may be attributed to increased elasticity, leading to a reduction in the relative motion of the molecular chains. Therefore, only a small internal friction force needs to be overcome during the deformation process [33]. Furthermore, Figure 14 shows that the incorporation of WMA into RAB increased the value of the damping ratio. However, the increment gradually decreased with a decrease in temperature until an almost equal value was finally achieved. This implied that there was only slight improvement in the thermal crack resistance of RAB due to the WMA additive. Moreover, the effect of SK was found to be more prominent than that of LP especially at −24 °C. For example, the damping ratio increased from 0.4046 to 0.4673 for RAB containing 1, 2, and 3% LP, while the addition of 0.4, 0.6, and 0.8% SK led to an increase in the damping ratio from 0.4355 to 0.5411, which was 15% greater than that of L-RABs. It was also found that this result of optimal S0.6-RAB was significantly larger than that of optimal L3-RAB by 16%. Therefore, the performance of S-RABs was better than that of L-RABs.

#### 4.2.4. Dissipation Energy Ratio of Asphalt Binders

The dissipation energy ratio DER was used to reflect the degree of internal flow property and stress relaxation at low temperatures. The DER value was calculated using Equation (19), as shown in Figure 15. For any viscoelastic material, a higher value of DER indicates better low-temperature performance. Figure 15 demonstrates that the DER experienced a rapid decline when the temperature decreased from −12 °C to −18 °C. However, when the temperature decreased from −18 °C to −30 °C, the DER showed a slow decline. This was partly because the molecular motion energy decreased, thereby imprisoning the chain motion and forming a strong structure. Evidently, the elastic behavior exhibited a conspicuous increase, and the stress relaxation ability exhibited a significant decline following a decrease in temperature. Moreover, the RABs containing WMA additives possessed better low-temperature properties, especially for 0.6% SK and 3% LP, which is consistent with the conclusion of the damping ratio.

Overall, based on the detailed analysis of m; parameters a, $D\left(t\right)$, and $D^{\prime }\left(t\right)$; the damping ratio; and DER, it can be concluded that the simple fractional model can be used for comparing the results of experimental tests. Moreover, the WMA additives imparted a positive influence on the thermal anti-cracking ability, while the effect decreased with a decrease in temperature.

## 5. Conclusions

In this study, the influence of different types and contents of warm-mix additives on the low-temperature performance of recycled rubber asphalt binder (RAB) was evaluated using BBR test and the simple fractional viscoelastic model. Several parameters—thermal stress $\sigma \left(T\right)$, critical cracking temperature $T_{CR},$ Superpave specification parameters, the derivation of creep compliance $D^{\prime }\left(t\right),$ the damping ratio, and the dissipation energy ratio DER—were employed for evaluation. Based on the results of this study, the following conclusions can be drawn.Based on the BBR test results and the master curves of the relaxation modulus and $m_{E}$, it was estimated that the surfactant (SK) and viscosity reducer (LP) exhibited a positive effect on the low-temperature cracking resistance of RAB and S-RAB, which was better than that of L-RAB.Results of $\sigma \left(T\right),\;\;T_{CR}$, and the fractional model further proved that the content of WMA additives showed the decisive influence on the low-temperature properties of RAB. Two WMA additives led to a decrease in the critical cracking temperature $T_{CR}$ and the thermal stress $\sigma \left(T\right)$ of RAB, which decreased linearly with the increase in JN content, but the intermediate dosage of SK was the largest. The simple fractional model could suitably predict the creep compliance of W-RABs over a wide temperature range with only two parameters $\left(A,\;a\right)$ by fitting the BBR test results. The WMA additives improved the damping ratio, DER, and $D^{\prime }\left(t\right)$ of RAB, and the maximum improvement was observed for SK at dosage of 0.6%.The compound of RAB and SK (0.6 wt.%) modifier was recommended, which was found to be more suitable for use in asphalt pavements in extremely cold areas (such as northwest China). On the other hand, only a high dosage of LP (3 wt.%) could compensate for the weakness of L-RABs at low temperatures.


These results indicate that the WMA additives exhibit a minor positive effect on the low-temperature cracking properties of RAB, while the difference in dosage is significant. Undeniably, a lot more systematic explorations are still demanded for further extending the characterization of the microscopic properties of W-CRABs to explain the modification and aging mechanism of recycled rubber powder and WMA additives, which will be pursued in the future.

## Figures and Tables

**Figure 1 materials-15-04389-f001:**
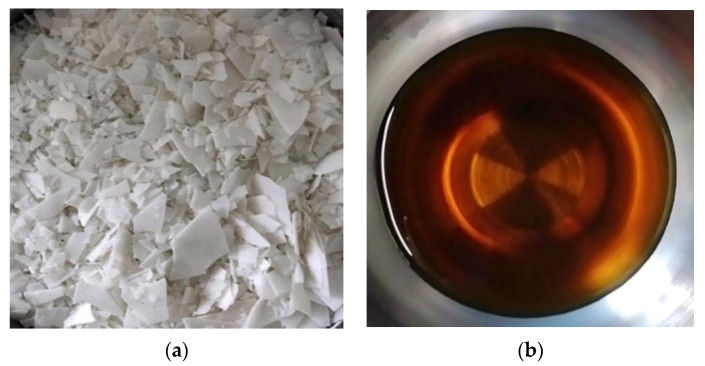
(**a**) LP-viscosity reducer additive; (**b**) SK-surfactant additive.

**Figure 2 materials-15-04389-f002:**
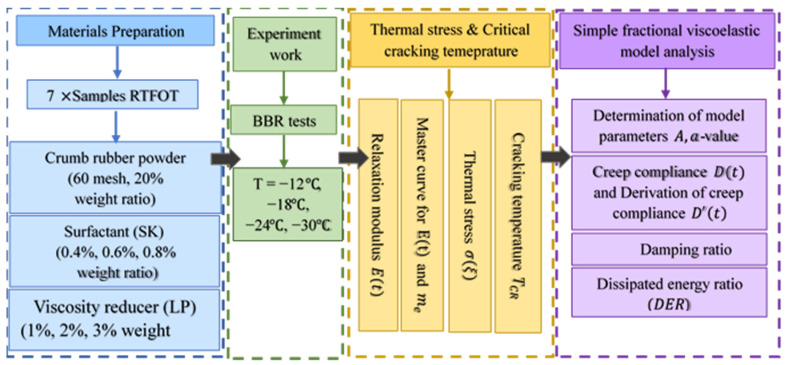
Schematic showing the research approach.

**Figure 3 materials-15-04389-f003:**
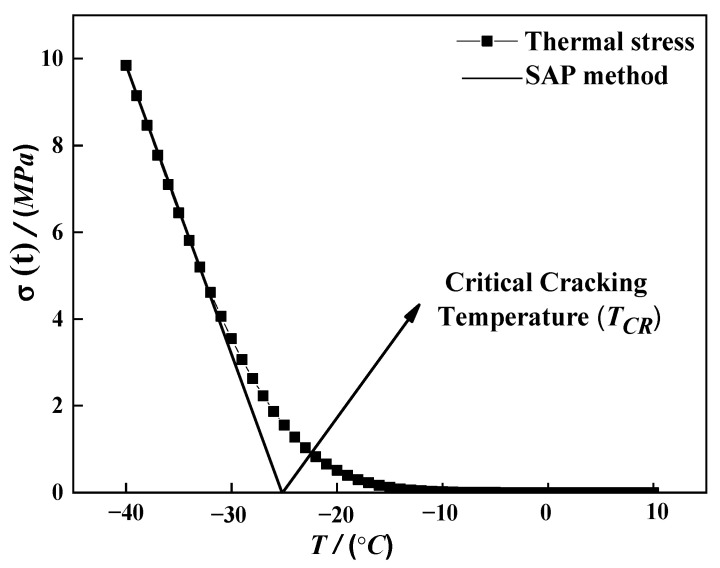
Single asymptotic procedure [20].

**Figure 4 materials-15-04389-f004:**
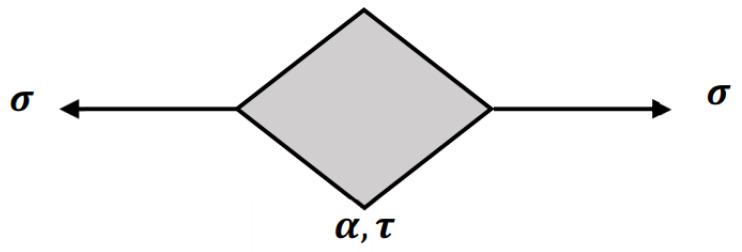
Simple fractional model viscoelastic element.

**Figure 5 materials-15-04389-f005:**
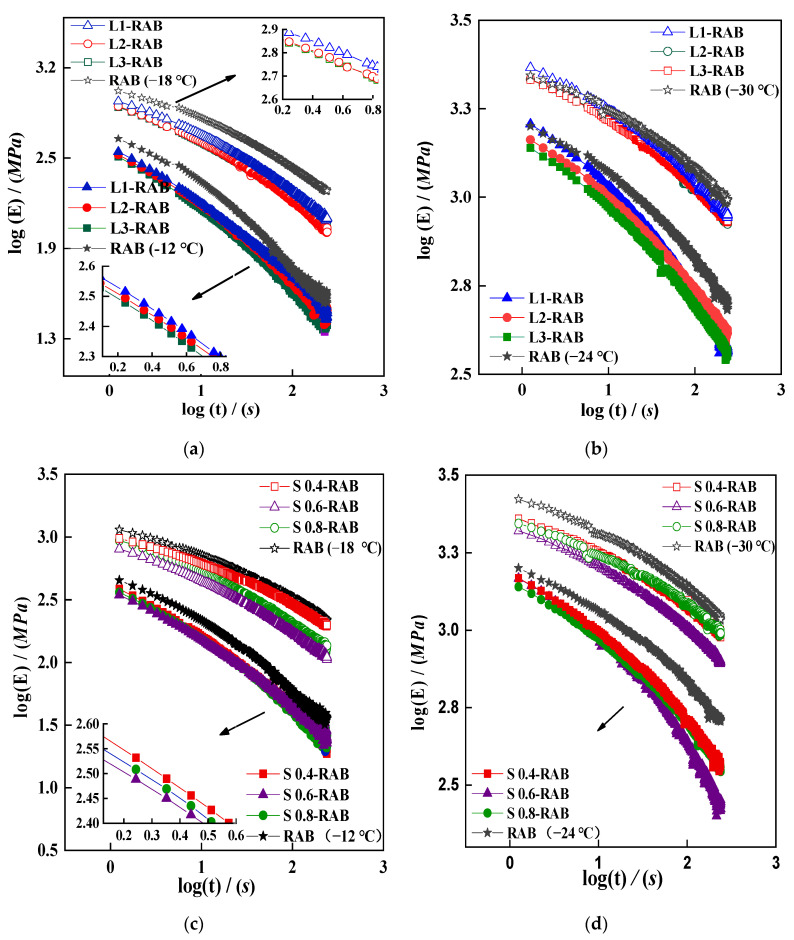
$E\left(t\right)$ curves of W-RABs at −12 °C, −18 °C, −24 °C, −30 °C. (**a**) L-RABs at −12 °C and −18 °C; (**b**) L-RABs at −24 °C and −30 °C; (**c**) S-RABs at −12 °C and −18 °C; (**d**) S-RABs at −24 °C and −30 °C.

**Figure 6 materials-15-04389-f006:**
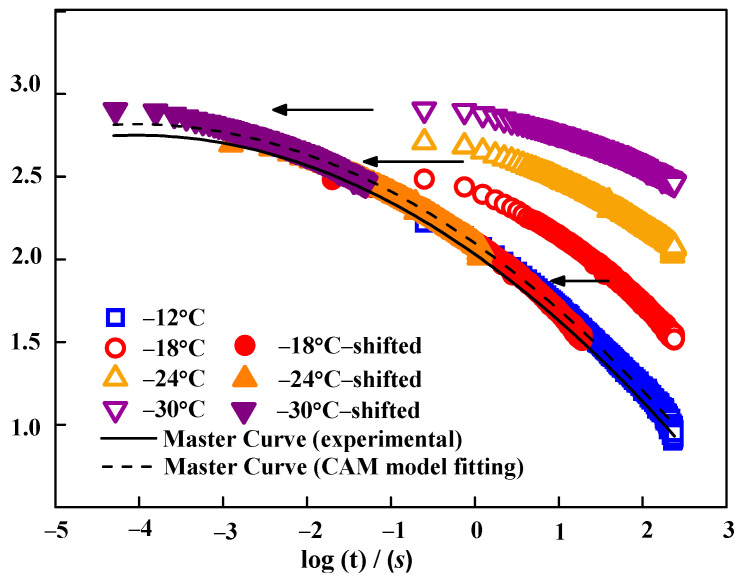
The $E\left(t\right)$ movement process for L3-RAB.

**Figure 7 materials-15-04389-f007:**
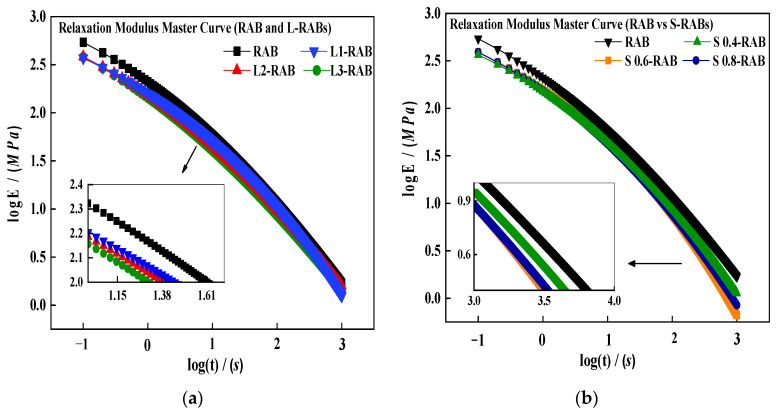
The master curves of $E\left(t\right)$ for W-RABs at a reference temperature of − 12 °C. (**a**) RAB and L-RABs; (**b**) RAB and S-RABs.

**Figure 8 materials-15-04389-f008:**
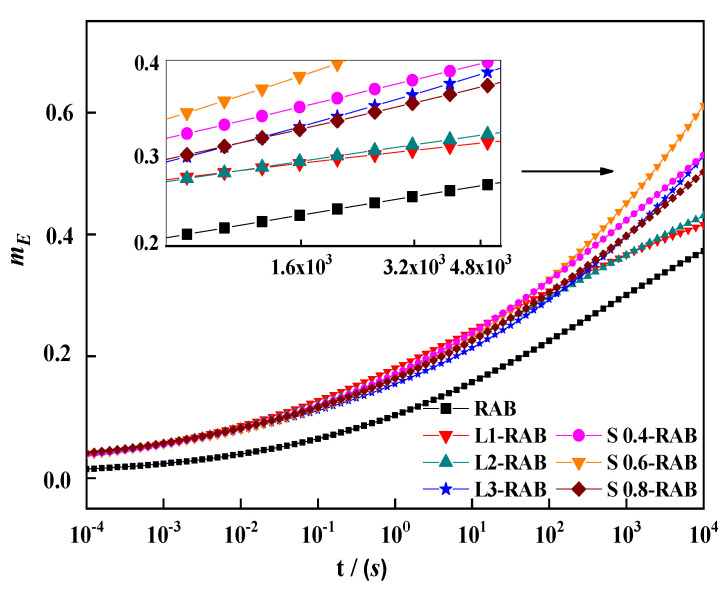
The master curves of $m_{E}$ for W-RABs at a reference temperature of − 12 °C.

**Figure 9 materials-15-04389-f009:**
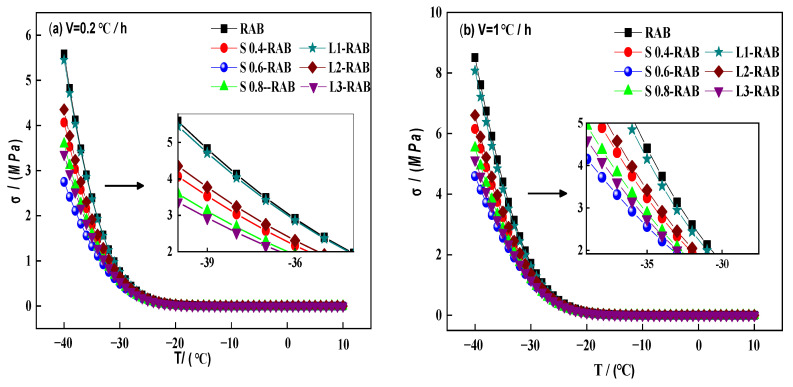
$\sigma \left(T\right)$ for W-RABs (**a**) V = 0.2 °C/h; (**b**) V = 1 °C/h; (**c**) V = 5 °C/h; and (**d**) V = 20 °C/h.

**Figure 10 materials-15-04389-f010:**
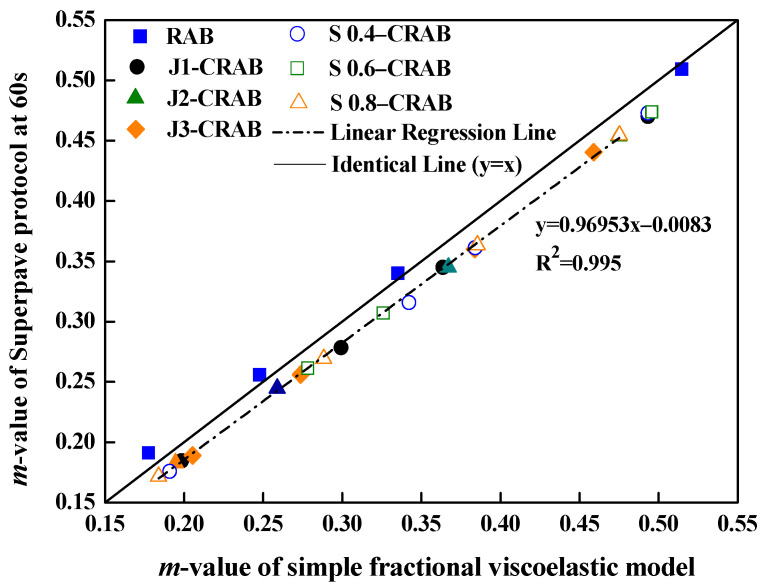
m-Values of simple fractional viscoelastic model versus the Superpave protocol.

**Figure 11 materials-15-04389-f011:**
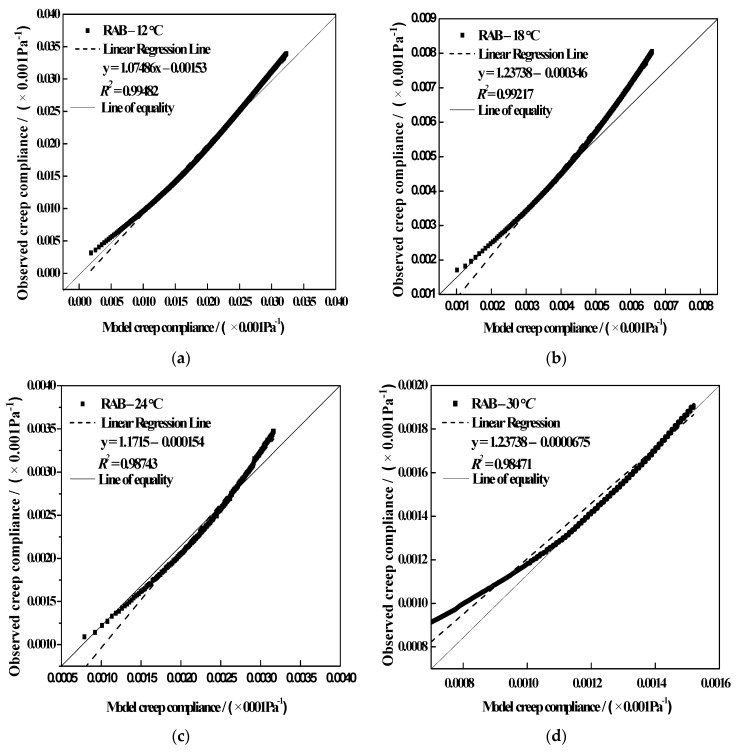
$D\left(t\right)$ values of simple fractional viscoelastic model versus experimental results for RAB at (**a**) −12 °C, (**b**) −18 °C, (**c**) −24 °C, and (**d**) −30 °C.

**Figure 12 materials-15-04389-f012:**
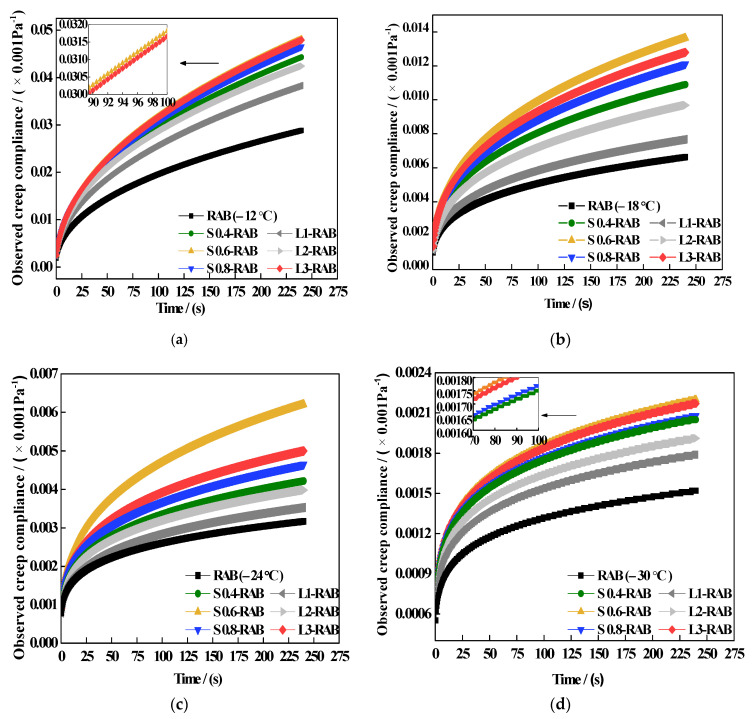
$D\left(t\right)$ values of simple fractional viscoelastic model for W-RABs at (**a**) −12 °C, (**b**) −18 °C, (**c**) −30 °C, and (**d**) −30 °C.

**Figure 13 materials-15-04389-f013:**
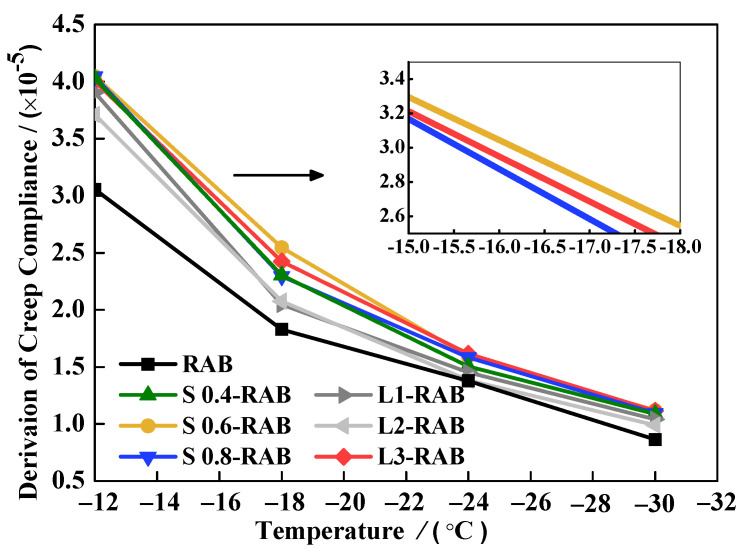
$D^{\prime }\left(t\right)$ values for W-RABs at 60 s.

**Figure 14 materials-15-04389-f014:**
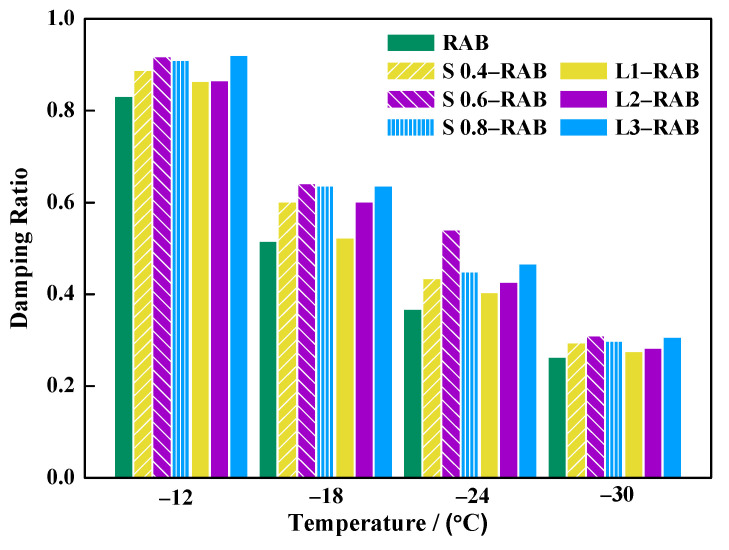
Results of the damping ratio based on the simple fractional model.

**Figure 15 materials-15-04389-f015:**
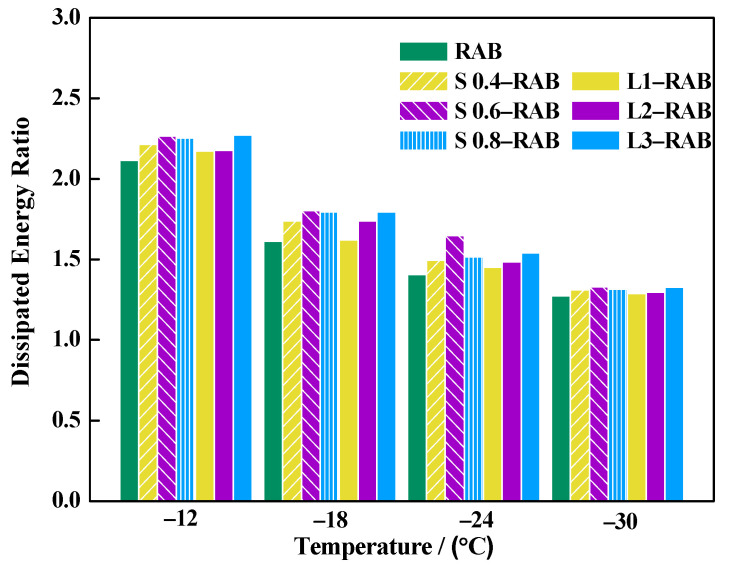
Results of DER based on the simple fractional model.

**Table 1 materials-15-04389-t001:** Physical properties of CR.

Index	60 Mesh	Technical Index
Density/(g cm^−3^)	1.10	1.1–1.3
Heating loss/%	0.6	≤1
Ash content/%	6.0	≤8
Iron content/%	0.021	≤0.3
Fiber content/%	0.4	<1

**Table 2 materials-15-04389-t002:** Nomenclature of asphalt binders.

Samples	Definition
RAB	Recycled crumb rubber-modified asphalt binder
W-RAB	Warm-mixed crumb rubber-modified asphalt binder
L-RAB	RAB with viscosity reducer additive (LP)
S-RAB	RAB with surfactant additive (SK)
L1-RAB	RAB with 1% LP
L2-RAB	RAB with 2% LP
L3-RAB	RAB with 3% LP
S0.4-RAB	RAB with 0.4% SK
S0.6-RAB	RAB with 0.6% SK
S0.8-RAB	RAB with 0.8% SK

**Table 3 materials-15-04389-t003:** $T_{CR}$ for W-RABs.

$v/\left({}^{\circ}C\cdot h^{-1}\right)$	$T_{CR}/{}^{\circ}C$						
	RAB	S0.4-RAB	S0.6-RAB	S0.8-RAB	L1-RAB	L2-RAB	L3-RAB
0.2	−27.4653	−29.1739	−29.3569	−29.221	−28.6015	−29.1372	−29.3123
1	−27.2784	−27.9725	−28.1871	−28.0457	−27.7018	−27.9540	−28.1432
5	−25.4904	−26.4228	−26.5978	−25.4904	−26.2628	−26.3585	−26.5086
20	−23.4429	−24.3760	−24.7374	−24.6937	−24.1650	−24.3600	−24.7122

**Table 4 materials-15-04389-t004:** Simple fractional viscoelastic model parameter for all asphalt binders.

Samples	−12 °C		−18 °C		−24 °C		−30 °C	
	A	a	A	a	A	a	A	a
**RAB**	2.56 × 10^−3^	0.4417	0.25 × 10^−3^	0.3038	9.23 × 10^−4^	0.225	6.18 × 10^−4^	0.1641
**L1-RAB**	3.18 × 10^−3^	0.4540	1.43 × 10^−3^	0.3072	9.23 × 10^−4^	0.2448	6.99 × 10^−4^	0.1717
**L2-RAB**	3.51 × 10^−3^	0.4545	1.46 × 10^−3^	0.3450	9.72 × 10^−4^	0.2573	7.29 × 10^−4^	0.1758
**L3-RAB**	3.57 × 10^−3^	0.4739	1.77 × 10^−3^	0.3613	1.09 × 10^−3^	0.2783	7.66 × 10^−4^	0.1902
**S0.4-CRAB**	3.51 × 10^−3^	0.4625	1.65 × 10^−3^	0.3450	1.01 × 10^−3^	0.2615	7.54 × 10^−4^	0.1828
**S0.6-CRAB**	3.60 × 10^−3^	0.4730	1.86 × 10^−3^	0.3636	1.10 × 10^−3^	0.3157	7.68 × 10^−4^	0.1919
**S0.8-CRAB**	3.54 × 10^−3^	0.4702	1.67 × 10^−3^	0.3613	1.06 × 10^−3^	0.2693	7.56 × 10^−4^	0.1847

## Data Availability

Not applicable.

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
