# Peer review of "Cracking Resistance of Recycled Rubber Asphalt Binder Composed of Warm-Mix Additives"

_materials, 2022, doi:10.3390/ma15134389_

Round 1

Reviewer 1 Report

The manuscript presents the analysis of the cracking resistance of recycled rubber asphalt binder composed of warm mix additives.

The paper is well structured and correctly developed.

The entire article is correct. The experiment was properly selected. The results are analyzed and described understandably. The conclusions require expansion. They are too general.

The experiment presents the analysis of two cases of additives (LP, SK) with different percentages (for LP 1.0; 2.0 and 3.0%; for SK 0.4, 0.6 and 0.8%). The conclusions do not detail how the additives affect the CR properties. Additionally, the recycled rubber asphalt aspect was completely omitted from the conclusions.

Please complete the applications with the impact of the allowances.

Reviewer 2 Report

Overall, this manuscript deals with a good inquest on binders cracking resistance using asphalt rubber and warm additives. The following comments are to be addressed carefully to improve the work's quality.

Abstract:

1. In the first paragraph, line 9, "...but the recycled rubber asphalt binder (RAB) causes toxic and carbon emissions, " is unnecessary, and this statement could be incorrect.

2. The authors have to make clear in the Abstract what the main objective of the research is.

Background

1. Line 32: "Low-temperature cracking is a major distress for asphalt pavement". Low-temperature cracking is one of them.

The low-temperature analysis is irrelevant in many tropical and subtropical countries, and there is other significant distress such as fatigue (intermediary temperatures), cracking propagation and permanent deformation.

2. There are problems in ordering references throughout the text (in square brackets). Please check and correct.

Research objective:

Still, it was not possible to identify the main objective of the research. This item presents a brief of the methodology procedures and not the main objective.

General

I think that when the pdf document was built, occurred problems. Many parts of the figures are missing, and the items are not separated adequately. Please verify.

Theoretical Basis

A critical concern is that the authors have merely presented the theoretical basis. Further discussion on the survey procedure needs to be included, and an in-depth analysis must be performed. Maybe this item belongs to the Background. Please include the proper reference in Figure 3.

Results

1. Please provide improvements in the figures, the quality is not proper, and the font is different in some of them.

2. Discussion is missing in this work

Conclusions

The conclusion should contain the following items:

1. Contribution to the body of knowledge,

2. Contribution of the study to the practice,

3. Limitation of the research,

4. Possible implications,

5. Further studies recommendations.

Round 2

Reviewer 2 Report

The corrections requests were answered, and now, that paper can be published. Thanks to the authors.